# Evolutionary Fate of the Opine Synthesis Genes in the *Arachis* L. Genomes

**DOI:** 10.3390/biology13080601

**Published:** 2024-08-09

**Authors:** Olesja D. Bogomaz, Victoria D. Bemova, Nikita A. Mirgorodskii, Tatiana V. Matveeva

**Affiliations:** 1Faculty of Bioengineering and Bioinformatics, Moscow State University, Moscow 119991, Russia; olesjabogomaz@yandex.ru; 2Department of Oil and Fibre Crops, N.I. Vavilov All-Russian Institute of Plant Genetic Resources (VIR), Saint Petersburg 190031, Russia; 3Field of Study Plant Genetics and Biotechnology, University “Sirius” Krasnodar Region, Federal Territory, Sirius 354340, Russia; 4Department of Genetic and Biotechnology, St. Petersburg State University, Saint Petersburg 199034, Russia

**Keywords:** naturally transgenic plants, *Arachis* L., cT-DNA, cucumopine synthase

## Abstract

**Simple Summary:**

The ancestors of modern peanuts underwent genetic transformation under natural conditions millions of years ago to acquire genes for opine synthesis. Eventually, the genes for the agropine synthesis pathway were lost in most of the studied samples or remained in the form of small fragments. Conversely, the genes for cucumopine synthesis remained intact and functional in most species, probably due to being under a regime of stabilizing selection. The variability of natural transgenes can be used in phylogenetic studies.

**Abstract:**

Naturally transgenic plants are plants that have undergone *Agrobacterium*-mediated transformation under natural conditions without human involvement. Among *Arachis hypogaea* L., *A. duranensis* Krapov. & W.C. Greg, *A*. *ipaensis* Krapov. & W.C. Greg, *A. monticola* Krapov. & Rigoni, and *A. stenosperma* Krapov. & W.C. Greg are known to contain sequences derived from the T-DNA of “*Agrobacterium*”. In the present study, using molecular genetics and bioinformatic methods, we characterized natural transgenes in 18 new species from six sections of the genus *Arachis*. We found that small fragments of genes for enzymes of the agropine synthesis pathway were preserved only in some of the studied samples and were lost in the majority of the species during evolution. At the same time, genes, similar to cucumopine synthases (*cus*-like), remained intact in almost all of the investigated species. In cultivated peanuts, they are expressed in a tissue-specific manner. We demonstrated the intraspecific variability of the structure and expression of the *cus*-like gene in cultivated peanuts. The described diversity of gene sequences horizontally transferred from *Agrobacterium* to plants helps to shed light on the phylogeny of species of the genus *Arachis* and track possible hybridization events. Data on the ability of certain species to hybridize are useful for planning breeding schemes aimed at transferring valuable traits from wild species into cultivated peanuts.

## 1. Introduction

*Agrobacterium*-mediated transformation, now referred to as *Rhizobium*-mediated transformation, is a commonly used method for obtaining genetically modified crops. This involves the transfer of a fragment of the bacterium’s plasmid, called T-DNA (transferred DNA), into the plant’s chromosome. The T-DNA genes contain eukaryotic promoters that allow for their expression in the plant. This expression leads to the development of crown gall or hairy root diseases, as well as the production of opines, which serve as a source of nutrition for the bacterium. Crown galls and hairy roots are transgenic tissues that form on non-transgenic plants. Over time, scientists have learned how to replace the internal T-DNA sequence with the genes of interest, and these genes can be transferred into plant cells, resulting in the regeneration of whole transgenic plants [1]. At the same time, some plant species have undergone transformation by *Agrobacterium* or *Rhizobium* naturally, without any human involvement. They were first described inside the genus *Nicotiana* and are referred to as naturally transgenic or natural genetically modified organisms (nGMOs) [2,3,4]. These plants contain sequences in their genomes called cellular T-DNAs (cT-DNAs), which are homologous to “agrobacterial” T-DNAs. These cT-DNAs are passed down to their offspring from generation to generation [5]. Since 2019, many nGMOs have been identified among various crop species, and most of them contain genes for opine synthesis in their genomes [6,7,8].

One example of such nGMOs is a group of species that includes cultivated peanuts (*Arachis hypogaea* L.) and their ancestors: *A. duranensis* Krapov. & W.C. Greg and *A. ipaensis* Krapov. & W.C. Greg. This group also includes two other species: *A. monticola* Krapov. & Rigoni and *A. stenosperma* Krapov. & W.C. Greg [6,9]. Cultivated peanuts and *A. monticola* are tetraploid and contain ancestral genomes known as A and B. Genome A is also present in *A. duranensis* and *A. stenosperma*, while *A. ipaensis* contains genome B [10]. The homolog of the *Agrobacterium* or *Rhizobium* cucumopine synthase gene (*cus*) [11,12] has been found in the genomes of all these *Arachis* species [6]. In this text, we will refer to this homolog as the *cus*-like gene. In the A genome of these species, the transgene is intact, whereas in the B genome, it is truncated. In addition to the mutated *cus*-like gene, remnants of the *mas’2* gene, which is involved in mannopine synthesis [13,14,15], are also found in the B genome of *A. ipaensis* and related tetraploid species [6]. All of the above species belong to the *Arachis* section [16]. In addition to it, the genus includes the sections *Erectoides* (genome E), *Extranervosae* (genome EX), *Procumbentes* (genome PR), *Caulorrhizae* (genome C), *Heteranthae* (genome H), *Rhizomatosae* (genome R), *Trierectoides* (genome TE), and *Triseminatae* (genome T) [16]. Until recently, species from these sections had not been examined for the presence of cT-DNA. Our research aims to fill this knowledge gap and shed light on the widespread occurrence of naturally occurring GMOs among cultivated *Arachis* species and their close relatives. The idea that transgenic plants have been consumed as food throughout history is particularly significant in supporting GMOs in countries where their cultivation is prohibited. Moreover, the examination of polymorphisms in recently acquired cT-DNA sequences can help to resolve contentious issues in peanut phylogeny. These sequences offer numerous advantages for investigating the origins and evolution of nGMO species. cT-DNAs are well-defined, highly specific, and easily identifiable DNA fragments that are distinct from plant sequences. Integration at a specific chromosomal site in the plant genome indicates a founder event, thereby establishing a clear starting point for a new clade. Additionally, cT-DNAs can be of sufficient size and age to produce a range of variants, which allow for the construction of phylogenetic trees. By analyzing the evolutionary trajectory of transgenes in a closely related group of species, researchers can gain insights into how they evolve [15]. Phasing T-DNA alleles allows for the reconstruction of more accurate phylogenetic relationships and the detection of hybridization events during plant species evolution. These methods have been effectively applied in studying the genera *Camellia* and *Vaccinium* [17,18]. In this study, the same approaches were used to characterize the natural transgenes of *Arachis* and their contribution to phylogenetic research.

## 2. Materials and Methods

### 2.1. Plant Material

*A. hypogaea* lines from the VIR collection were used to study the biodiversity and expression of the *cus*-like gene. The characteristics of the lines are presented in Appendix A. The NCBI SRA database was used to search for T-DNA in other species.

### 2.2. DNA and RNA Isolation

High-molecular-weight DNA was extracted from fresh 2-week-old seedlings of *A. hypogaea* cultivated aseptically in vitro, according to the CTAB protocol for DNA isolation from plant tissues [19].

Three-week-old aseptic *A. hypogaea* plants were used for RNA extraction (three plants per line). All plants were approximately the same size and were in equally healthy conditions in terms of leaf color and the absence of signs of wilting. For each plant, a total of three RNA samples were isolated from the leaves, roots, and internodes. Immediately after collection, the samples were flash-frozen and homogenized in liquid nitrogen. RNA was extracted from leaves, roots, and internodes using an RNeasy Plant Mini Kit (Qiagen, Hilden, Germany) according to the manufacturer’s instructions. The purity and quantity of RNA in the samples were measured using the spectrophotometer NanoDrop 2000/2000c (Thermo Fisher, Waltham, MA, USA).

### 2.3. PCR

The *cus*-like gene sequence was PCR-amplified using gene-specific primers (Appendix A). The amplification reactions (40 μL) contained a DNA sample, 20 μL of DreamTaq PCR Master Mix (2×) (Thermo Fisher, Waltham, MA, USA), and 10 pmol of each primer. Primers are listed in Appendix A. PCR was performed using a “Tertsyk” DNA amplifier (“DNA technology”, Moscow, Russia) according to the following program: 5 min at 94 °C, 35 cycles of 15 s at 94 °C, 30 s at 55 °C, and 60 s at 72 °C, followed by 10 min at 72 °C. PCR products were separated on an agarose gel in 1× TBE buffer, visualized using GelDoc Go (BioRad, Hercules, CA, USA), and used for further Sanger DNA sequencing.

### 2.4. DNA Sequencing

PCR fragments were sequenced using the Sanger method and the BrilliantDye™ Terminator (v3.1) Cycle Sequencing Kit (NimaGen, Nijmegen, The Netherlands). Then, sequencing mixtures were analyzed using an ABI Prism 3500 xl sequencer (Applied Biosystems, Waltham, MA, USA).

### 2.5. Reverse Transcription and Real-Time qPCR

Before cDNA synthesis, the amounts of RNA in different tissues were normalized. For each sample, cDNA was synthesized with 330 ng of total RNA using an iScript cDNA Synthesis Kit (Bio-Rad, Hercules, CA, USA) according to the manufacturer’s instructions. The cDNA synthesis reaction (20 μL) was performed in 0.2 mL tubes on a Bio-Rad C1000 Thermal Cycler apparatus as follows: 5 min at 25 °C, 25 min at 46 °C, and 1 min at 95 °C. Real-time qPCR reactions (20 μL) were performed in optical 8-tube strips with a Bio-Rad CFX96 apparatus using an iTaq Universal SYBR Green Supermix 2× ((both from Bio-Rad), Hercules, CA, USA) and gene-specific primers (Appendix A). The A. hypogaea glyceraldehyde 3-phosphate dehydrogenase gene (gapdh) was used as a reference for the equalization of RNA levels. All primers were synthesized by Evrogen (Moscow, Russia). PCR was performed according to the following program: 5 min at 94 °C, 30 cycles of 20 s at 94 °C, 20 s at 55 °C, and 30 s at 72 °C, followed by 10 min at 72 °C.

Quantification was performed in triplicate. Relative expression levels were calculated using the 2−ΔΔCT method [20] for each sample, and average values for three plants were calculated.

### 2.6. Allele Reconstruction from SRA Data

The NCBI SRA database was used to search for T-DNA in the *Arachis* genus. The search was performed using BLAST against a reference (OL840910.1). Resulting reads were aligned to the reference using BWA 0.7.17 [21]. The processing of SAM files was performed using SAMtools 1.7 [22]. The alignment visualization for the ploidy estimation was performed using IGV 2.12.3 [23]. Allele phasing was performed using variant calling with GATK 4.2 [24], followed by WhatsHap 1.0 [25]. In cases of small coverage, alleles were phased manually, and some were partially successful. Based on the assumption that SNPs are artificial, their presence was verified using other samples.

### 2.7. Phylogenetic Analysis

Nucleotide sequences of separate alleles of the *cus*-like gene (listed in Appendix A) were aligned using MUSCLE v.5, with default parameters [26]. The evolutionary history was inferred using the maximum-likelihood method and the Kimura 1980 model [27] using PhyML 3.0 [28] with smart model selection [29]. The operation of this tool included selecting the best model using the BIC criterion. The initial trees for heuristic search in the ML method were obtained automatically using the BioNJ algorithm. To test the support for the topology of the tree, 100 bootstrap replicates were obtained [30]. The final tree was visualized in MEGA 11. Branches corresponding to partitions reproduced in less than 70% of bootstrap replicates were collapsed. The percentage of replicate trees in which the associated taxa clustered together were shown next to the branches [31].

## 3. Results

### 3.1. Naturally Transgenic Species of the Arachis Genus

Until recently, cT-DNA has only been described in cultivated peanuts (*A. hypogaea*), its ancestors *A. duranensis* and *A. ipaensis*, and in closely related *A. monticola* and *A. stenosperma*. In the framework of this study, using the BLAST algorithm, cT-DNA sequences were described in the WGS data for six accessions of *A. hypogaea*, five accessions of *A. duranensis*, four accessions of *A. ipaensis*, and, finally, one accession each of *A. monticola*, *A. stenosperma*, and A. *cardenasii*. Sequences homologous to the *cus* gene were present in all of these genomes and were located on chromosome 8 in a common integration site, indicating that all these sequences were the result of a single transformation event. In addition, the 8th chromosome of the species with the B genome contained remnants of the *mas2*′ gene, but they were located in different sites. *A. cardenasii*, in addition to an intact copy of the *cus*-like gene, contained remnants of the *mas1′* gene in different localization sites as a result of an independent transformation event.

In summary, our search for homologs of the opine biosynthesis genes in the SRA database revealed new species of naturally transgenic plants in the *Arachis* genus. They contain sequences homologous to *cus*, *mas1′*, *mas2′*, and *ags* genes (Table 1).

Natural GMOs were found in the sections *Arachis*, *Erectoides*, *Extranervosae*, *Procumbentes*, *Caulorrhizae*, and *Heteranthae*, indicating that their common ancestor was transformed before these sections diverged. Genomic data for different species varied in terms of coverage quality. In some cases, the presence of a transgene could be inferred from single reads. Sequences with good coverage were further used to assemble full-length genes. The *cus*-like gene was found in all species, as listed in Table 1. In addition, the gene was sequenced in eight lines of peanuts from the VIR collection. In SRA data of several species, we also found remains of *mas1′*, *mas2′*, and *ags*-like genes.

### 3.2. Intra- and Interspecific Variability of the Cus-like Gene

To assess the intraspecific variability of the *cus*-like gene, we used sequences of alleles reconstructed from SRA reads or Sanger sequencing data of independent samples of *A. hypogaea* according to procedures previously described for the *Vaccinium species* [17]. The cultivated peanuts included in the study consisted of eight lines from the VIR collection and sequences from the SRA database for 19 genotypes. We reconstructed 29 *cus*-like gene variants from the A genome, which we will refer to as alleles. These alleles corresponded to 23 amino acid sequences. Surprisingly, even though peanuts are self-pollinating, there was a high proportion of heterozygotes for the *cus*-like gene. The reason for this phenomenon is still unclear, due to the lack of sufficient data. It could be the result of hybridization and selection in favor of heterozygotes, or there may be a probability of having two copies of the gene organized as tandem repeats that mutate independently. The variability of the transgene copy number can be illustrated using WGS data for different varieties of the diploid species *A. duranensis*. Thus, there are two copies of the transgene in the genome assembly of cultivar PI475845. At the same time, each of the cultivars, K30060 and V14167, contain one copy of the transgene.

The frequencies of the different alleles were not the same. The most common allele, designated as A, was found in 12 genotypes from both subspecies. When analyzing the amino acid sequence of the gene product, taking into account the degeneracy of the genetic code, allele A was found in 13 lines of cultivated peanuts and the ancestral form of *A. duranensis*. Alleles B and C were found in three accessions each, while D was found in two accessions. The remaining alleles were unique. Only three of the twenty-nine *cus*-like alleles in the A genome contained mutations inconsistent with function (Appendix A). The rest can be considered intact. It is worth noting that we found common alleles in both the SRA assembly data and the Sanger sequencing results of samples from the VIR collection, indicating consistent results from these methods. The alleles from the B genome of cultivated peanuts were all truncated and unique. The presence of the common intact alleles from the A genome in half of the studied genotypes of *A. hypogaea* and ancestral *A. duranensis* suggests that there is stabilizing selection in favor of the intact allele. However, such selection is absent for the alleles from the B genome, causing them to accumulate mutations. All reconstructed sequences of the *cus*-like genes from other species can also be considered intact, except for one allele of *A. hoehnei*.

Since not all alleles of the gene under study were intact, their phylogenetic relationship was reconstructed based on nucleotide sequences. The phylogenetic analysis showed two distinct clades in the phylogenetic tree (Figure 1A), corresponding to the A and B genomes of cultivated peanuts. Clade β included sequences from *A. ipaensis* and the B genome of *A. hypogaea* and *A. monticola*. Clade α was more complex, including sequences from the A genome of *A. hypogaea*, *A. monticola*, *A. duranensis*, and *A. paraguariensis* (E genome). The sequence closest to clade α was obtained from *A. valida* with the B genome. Both sequences of *A. stenophylla* with the E genome and *A. pintoi* with the C genome were closer to clade β. The two sequences of *A. helodes* (genome A) clustered together. Clade γ included all analyzed alleles of *cus*-like genes from the PR genomes and *A.villosa* with the A genome. The sequences from two plants of *A. diogoi* and one plant of *A. hoechnei* formed clade ε, while two other alleles from these species were located outside of this clade. This suggests that interspecific hybridization may have occurred during the course of their evolution. The lower diversity of sequences from the A genome of cultivated peanuts may be due to gene functionality and stabilizing selection in favor of intact alleles, indicating potential gene expression. Therefore, we assessed the expressions in several peanut lines from the VIR collection.

### 3.3. Expression of the Cus-like Gene

The *cus*-like gene was expressed in cultivated peanuts (Figure 1B). Gene expression levels varied between lines during the juvenile stage. The analysis included two lines with allele A and two lines with allele B. In both cases, we observed lines with the same gene sequence but differing expression levels. This indicates that they differed in the genetic environment but not in the structure of the coding sequence of the transgene. In addition, there was a tendency for higher transgene expression in roots than in other plant organs, which is consistent with literature data for other species [32].

## 4. Discussion

In the presented work, we described cellular T-DNAs in the genomes of *Arachis* species from the sections *Arachis*, *Erectoides*, *Extranervosae*, *Procumbentes*, *Caulorrhizae*, and *Heteranthae*, supporting the idea of their monophyletic origin [31]. In all of the studied genomes, sequences homologous to cucumopine synthase were identified.

Among the currently known strains of *Agrobacterium*, the strain that has a gene with the highest similarity to the *cus*-like sequence of *Arachis* is *A. salinitolerans* LMG 29287 (JAPZLN010000006.1). *A. salinitolerans* is an aerobe and is a mesophilic, Gram-negative bacterium that forms circular colonies and was isolated from surface-sterilized effective root nodules of *Sesbania cannabina*, demonstrating the fine line between parasitism and mutualism [33].

In addition to the *cus*-like gene, species with genome B contained remains of the *mas2*′ gene, species with genome PR contained remains of the *mas1′* gene, *A. macedoi* contained remains of the *ags* gene, and *A. pusilla* contained remains of both the *mas2*′ and *ags* genes. All of these genes encode enzymes belonging to the same biosynthetic pathway. They catalyze a chain of reactions leading to the synthesis of agropine and are found clustered together in the same agrobacterial T-DNA [34]. The common ancestor of the *Arachis* species was probably transformed by a strain containing all three genes, but at some point, these genes ceased to provide selective advantages to their owners, accumulated mutations, and were lost, remaining in the form of separate fragments in representatives of different clades. One may speculate that the same fate awaits transgenes introduced into plants by humans in the absence of selection in their favor.

A completely different picture is typical for the cucumopine synthase gene. It has remained intact in most of the studied species. In cultivated peanuts, among the 29 described alleles of the gene, only 3 contained mutations that were incompatible with the function. The cultivated peanut is divided into two subspecies, *hypogaea* and *fastigiata* [35,36,37]. The most common allele A of the *cus*-like gene of *A. hypogaea* was found in representatives of both subspecies, as well as in *A. duranensis*, confirming the close relationship of these species. However, this allele has not yet been identified in *A. monticola*. Alleles of the *cus*-like gene of *A. monticola* were distributed among distinct subclades within clade α. A similar mosaic arrangement of *A. monticola* and *A. hypogaea* sequences on the tree was observed by Tian and co-authors during the chloroplast phylogenomic analysis of *Arachis* [38].

It is well accepted that *A. duranensis* and *A. ipaensis* are the progenitor species of *A. hypogaea* [36,39]. In recent years, the scientific community has accepted the idea of a common origin for *A. hypogaea* and *A. monticola*. However, there is no complete clarity regarding the rank of differences between them. Yin et al., based on a comparative analysis of NGS data, proposed that wild tetraploid *A. monticola* was formed by hybridization between two diploid species, *A. duranensis* and *A. ipaensis*, and then evolved into *A. hypogaea* by artificial selection [40]. Tian and co-authors [38] reported that *A. monticola* could be a weedy subspecies of cultivated peanuts, since *A. hypogaea* is able to hybridize with *A. monticola* and produce fertile hybrids. Thus, *A. monticola* may have been derived from a more ancient hybridization event, while the accessions of *A. hypogaea* may have evolved later. Our data are more consistent with the second hypothesis because the alternation of sequences of *A. hypogaea* and *A. monticola* within one clade on the tree is most logically explained through hybridization events and therefore the absence of a complete interspecific barrier.

There are several studies showing that cultivated peanuts may have been derived from more than two progenitor species, including such species as *A. diogoi*, *A. cardenasii*, and *A. batizocoi*, in addition to well-accepted ones [41,42,43,44,45]. Within our study, there is no evidence in favor of the participation of the species mentioned above in the formation of cultivated peanuts. However, it cannot be ruled out that the genetic material of *A. paraguariensis* was involved in the formation of at least some varieties of *A. hypogaea*. Data from other researchers support the close relationship between *A. paraguariensis* and *A. duranensis* [46]. The locations of other species on the tree do not agree well with the classical division of the genus into sections. For example, some alleles of *A. diogoi* and *A. hoechnei* form a common clade, while others are located outside of this clade. A similar pattern was previously observed by Tian and co-authors, who studied chloroplast sequences of *Arachis* species [38]. This can be explained by hybridization in the context of incomplete speciation.

Comparing our data with those of other researchers [38,44,45,46,47,48,49], it can be noted that the discrepancy in results between different studies is most likely due to the different breadth of material covered and different samples of representatives of the studied species. Thus, the origination and evolution of the *Arachis* species remain elusive, and it is extremely difficult to demarcate the boundaries of some peanut species, due to gene introgression, ancestral polymorphism, and various speciation rates in different species [46,47,48,49]. Further research necessitates the inclusion of additional lines, varieties, and species of *Arachis*. When examining a vast number of samples from different species, it is essential to employ cost-effective and efficient research methods for phylogenetic analyses. In addition to other techniques, incorporating a molecular marker based on a recently evolved sequence can provide a more comprehensive understanding of the phylogeny of *Arachis*.

Intact sequences of natural transgenes have now been described for many species [2,3,4,5,6,7,8,9]. Such sequences have attracted attention from the very beginning for studying their functions [50,51,52,53]. We also demonstrated the expression of the *cus*-like gene at the RNA level in this study. Our data show that expression can be considered tissue-specific, as the highest transcript content in most lines was observed in the roots. The same trend was noted earlier for tobacco [32]. It is interesting to note that the level of transgene expression varies among lines. Further studies on the functioning of natural transgenes using genetic material that contrasts in transgene expression will allow us to gain a better understanding of their function. Since it is known that opines can be secreted into the environment from plant cells [54,55] and can attract certain groups of microorganisms [56,57], it is logical to expect the influence of opines on the biological environment of natural GMOs. This could be an interesting area for further research. In addition, being derivatives of amino and keto acids [58], opines can affect protein metabolism, which also requires attention in future studies of this crop. Moreover, the fact that residents of South and North America, Africa, and Eurasia have been using GMOs with intact functional transgenes for centuries without harm to their health should alleviate the fears of modern society regarding the unnaturalness of GMOs. In laboratory conditions, people replicate the processes that have occurred and are occurring in nature. Therefore, it makes sense to test the effects of specific transgenes on humans and the environment rather than fearing transgenesis as a process.

## 5. Conclusions

Thus, in the present study, we characterized natural transgenes in *A. batizocoi*, *A. cardenasii*, *A. correntina*, *A. diogoi*, *A. duranensis*, *A. glandulifera*, *A. helodes*, *A. hoehnei*, *A. hypogaea*, *A. ipaensis*, *A. magna*, *A. monticola*, *A. stenosperma*, *A. trinitensis*, *A. valida*, *A. villosa*, *A. paraguariensis*, *A. stenophylla*, *A. macedoi*, *A. appressipila*, *A. pintoi*, *A. pusilla* Benth., and *A. rigonii*. We demonstrated the different evolutionary fates of the genes for agropine synthesis and cucumopine synthesis in *Arachis*. We showed the tissue-specific expression of cucumopine synthase in cultivated peanuts and characterized the intraspecific variability of the structure and the expression of the cus-like gene in *Arachis*. The described diversity of gene sequences horizontally transferred from *Agrobacterium* to plants helps shed light on the phylogeny of species of the genus *Arachis*.

## Figures and Tables

**Figure 1 biology-13-00601-f001:**
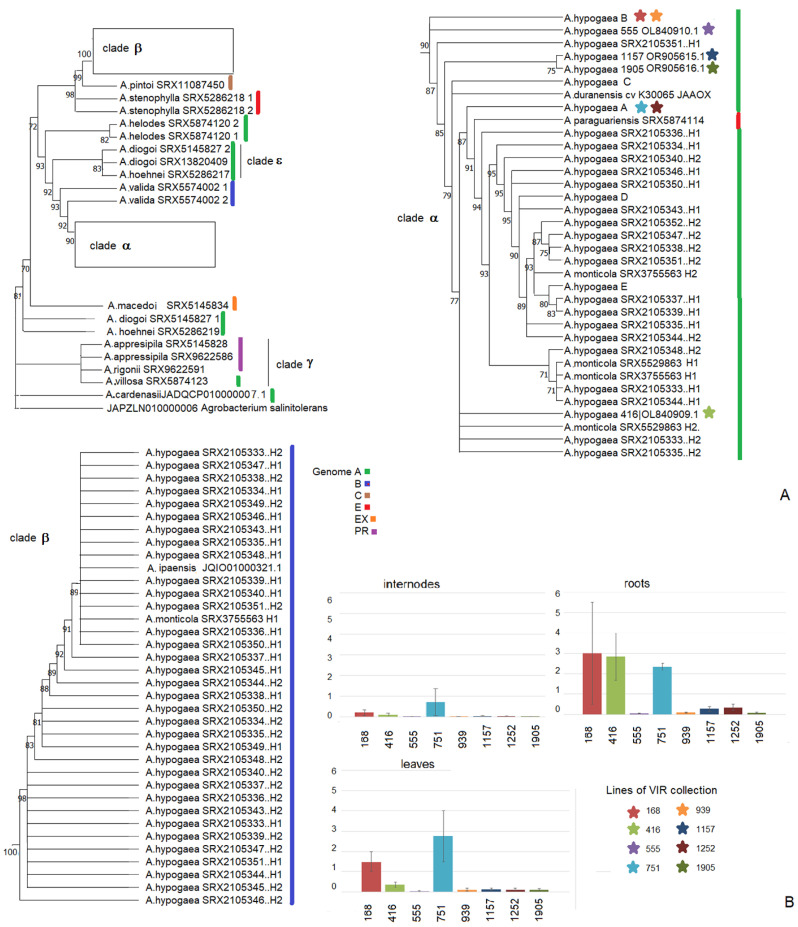
Phylogenetic relationships between *cus* -like sequences from different *Arachis* species and the intraspecific variability of *cus*-like gene expression in cultivated peanuts. (**A**) The evolutionary history of *Arachis* inferred based on the *cus*-like gene, using the maximum-likelihood method and the Kimura 1980 model. (**B**) Expression of the transgene in the lines of the VIR collection. The asterisks show the colors marking the different lines and their correspondence to alleles on the tree.

**Table 1 biology-13-00601-t001:** Natural transgenes of the *Arachis* species, based on NGS data.

Section [16]	Genome [16]	Species	NGS Data	Transgenes Found	SRR Accession Number(s)
*Arachis*	K	*A. batizocoi* Krapov. & W.C. Greg.	cT-DNA is present in the genome, but coverage is not sufficient for assembly	*cus*	SRX5286222, SRX5286221
	A	*A. cardenasii* Krapov. & W.C. Greg	cT-DNA was found in WGS data, and in new SRA data, coverage is sufficient to assemble full-length genes	*cus*	DRX231922, SRX5574006
	A	*A. correntina (Burkart)* Krapov. & W.C. Greg.	cT-DNA is present in the genome, but coverage is not sufficient for assembly	*cus*	SRX5874118
	A	*A. diogoi* Hoehne	cT-DNA is present in the genome, but coverage is not sufficient for assembly	*cus*	SRX1382040, SRX51458279
	A	*A. duranensis* Krapov. & W.C. Greg.	cT-DNA was described earlier, and in new SRA data, coverage is sufficient to assemble full-length genes	*cus*	SRX5574003
	D	*A. glandulifera* Stalker,	cT-DNA is present in the genome, but coverage is not sufficient for assembly	*cus*	SRX15154171
	A	*A. helodes* Mart. ex Krapov. & Rigoni	cT-DNA is present in the genome, but coverage is not sufficient for assembly	*cus*	SRX5874120
	A	*A. hoehnei* Krapov. & W.C. Greg.	cT-DNA is present in the genome, and coverage is sufficient to assemble a full-length gene	*cus*	SRX5286219
	AB	*A. hypogaea* L.	cT-DNA was described earlier, and in new SRA data, coverage is sufficient to assemble full-length genes	*cus*, *mas2′*	SRX2105338, SRX2105345, SRX2105341, SRX2105349, SRX2105334, SRX2105351, SRX2105347, SRX2105352, SRX2105333, SRX2105343, SRX2105344, SRX2105342, SRX2105348, SRX2105335, SRX2105337, SRX2105350, SRX2105340, SRX2105346, SRX2105336, SRX2105339, SRX4393155, SRX4393154, SRX4393152, SRX4393153
	B	*A. ipaensis* Krapov. & W.C. Greg.	cT-DNA was described earlier, and in new SRA data, coverage is sufficient to assemble full-length genes	*cus*, *mas2′*	
	B	*A. magna* Krapov. et al.	cT-DNA is present in the genome, but coverage is not sufficient for assembly	*cus*, *mas2′*	SRX5286220, SRX5573999
	AB	*A. monticola* Krapov. & Rigoni	cT-DNA was described earlier, and in new SRA data, coverage is sufficient to assemble full-length genes	*cus*, *mas2′*	SRX3755563, SRX3802089
	A	*A. stenosperma* Krapov. & W.C. Greg.	cT-DNA was described earlier, and in new SRA data, coverage is sufficient to assemble full-length genes	*cus*	SRX5286218
	F	*A. trinitensis* Krapov. & W.C. Greg.	cT-DNA is present in the genome, but coverage is not sufficient for assembly	*cus*	SRX15154170
	B	*A. valida* Krapov. & W.C. Greg.	cT-DNA is present in the genome, and coverage is sufficient to assemble a full-length gene	*cus*, *mas2′*	SRX5574002
	A	*A. villosa* Benth.	cT-DNA is present in the genome, and coverage is sufficient to assemble a full-length gene	*cus*	SRX5874123
*Erectoides*	E	*A.paraguariensis* Chodat & Hassl	cT-DNA is present in the genome, and coverage is sufficient to assemble a full-length gene	*cus*	SRX5874114
	E	*A. stenophylla* Krapov. & W.C. Greg.	cT-DNA is present in the genome, but coverage is not sufficient for assembly	*cus*	SRX5286218
*Extranervosae*	EX	*A. macedoi* Krapov. & W.C. Greg.	cT-DNA is present in the genome, and coverage is sufficient to assemble a full-length gene	*cus*, *ags*	SRX5145834
*Procumbentes*	PR	*A.appressipila* Krapov. & W. C. Greg.	cT-DNA is present in the genome, and coverage is sufficient to assemble a full-length gene	*cus*, *mas1′*	SRX5145828
	PR	*A. rigonii* Krapov. & W.C. Greg.	cT-DNA is present in the genome, and coverage is sufficient to assemble a full-length gene	*cus*, *mas1′*	SRX9622591
*Caulorrhizae*	C	*A. pintoi* Krapov. & W.C. Greg.	cT-DNA is present in the genome, and coverage is sufficient to assemble a full-length gene	*cus*	SRX11087450
*Heteranthae*	H	*A. pusilla* Benth.	cT-DNA is present in the genome, but coverage is not sufficient for assembly	*cus*, *mas2′*, *ags*	SRX5145826

## Data Availability

The sequences obtained during the work were deposited into the National Center for Biotechnology Information database (https://www.ncbi.nlm.nih.gov/ (accessed on 26 December 2023) with accession numbers OL840908.1–OL840912.1 and OR905615.1–OR905616.1).

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
