# Peer review of "Evolutionary Fate of the Opine Synthesis Genes in the Arachis L. Genomes"

_biology, 2024, doi:10.3390/biology13080601_

Round 1

Reviewer 1 Report (Previous Reviewer 1)

Comments and Suggestions for Authors

The authors have solved all the problems that I found in the previous version.

Reviewer 2 Report (Previous Reviewer 3)

Comments and Suggestions for Authors

This version is okay now.

This manuscript is a resubmission of an earlier submission. The following is a list of the peer review reports and author responses from that submission.

Round 1

Reviewer 1 Report

Comments and Suggestions for Authors

Review of "Evolutionary fate of the opine synthesis genes in the Arachis L. genomes" by O. D. Bogomaz et al.

This work describes the finding of genes derived from Agrobacterium-related bacteria in the genomes of plants of the Arachis genus. It provides sound evidence for both active and inactive opine synthesis genes being present in several new species of the genus.

The article may be, in my opinion, publishable, provided that several weak points are addressed.

Major problems:

1) The main problem that I detected concerns the phylogenetic analysis . I found that the conclusions of the authors do not follow from the data presented in Figure 1.

I will start by saying that Figure 1 is almost impossible to read. For example, what I imagine are the bootstrap values are barely visible when the manuscript is printed. I recommend presenting the tree not as a circle, but in the vertical, traditional way, so it would be possible to make the image bigger and increase the font of the letters.

I say "I imagine are bootstrap values" because the authors did not explicitly said which method they used to provide statistical support for the branches of the tree.

Regarding the conclusions, the alleles defined by the authors are not supported by the data shown in that tree. For such a qualitative partition of the tree, the authors should consider only those branches that have strong support. However, the branches that correspond to the alleles defined have support values that range from 60% to 99%. If indeed these values refer to bootstrap data, several of the alleles defined would lack any statistical support. Thus, the conclusions obtained may be incorrect.

I suggest that the authors repeat the phylogenetic analyses as follows:

a) Include an outgroup, which may be the closest bacterial relative to the Arachis sequences. This will allow them to root the tree, which may facilitate the definition of distinct groups within it.

b) The authors did not indicate whether they used DNA or protein sequences for this analysis. I deduced it is DNA because they say that they used Kimura's model. It is very likely that using protein sequences may provide better-supported trees.

c) I would definitely use IQTREE instead of PhyML for the maximum-likelihood analyses. IQTREE incorporates the possibility of testing which is the best model of sequence evolution. The authors did not explain why they used Kimura's model, when many other models are possible and potentially better. IQTREE compares a large set of alternative models and chooses the best one, given the data.

Only once a well-supported tree is obtained, would it be possible to define groups within it.

Other questions:

2) The authors should include in the Introduction all the data concerning the different ancestral genomes. They describe the A and B genomes, but not the others (C, E, PR...) that appear later in the text.

3) It would be very convenient to have a small figure showing the evolutionary relationships among the Arachis species discussed in the text. Otherwise, it is very difficult to follow the discussions on when the opine sequences appeared.

4) Also, it would be convenient to show in detail the data concerning the precise chromosomal locations of the sequences. How did the authors establish whether they are exactly the same in different species? If indeed they are able to map the sequences (line 145), why is it then impossible to establish whether the sequences come from a single gene or two genes in tandem? (as suggested on line 178).

5) Figure 1B: I do not understand why two alleles may come from a single branch in Figure 1A. Are they just replicates?. Also, it would be convenient to include in Figure 1B the names of the alleles, perhaps below the expression bars.

6) Were the first GMOs described in 2019? (line 47)

Comments on the Quality of English Language

The language must be improved. There are many minor mistakes (Table 1. "coverage is sufficient to assemble a full-length genes") and also phrases that cannot be understood (e. g. lines 269-271).

Author Response

The authors would like to express their gratitude to the reviewer for their time, valuable comments, and advice.
We have made revisions to the English language.

  1. The phylogenetic tree was constructed in accordance with the reviewer's comments.

Outgroup was included. However, we should understand that the gene from Agrobacterium is not an ideal outgroup. We still know little about the diversity of Agrobacterium. We don't know which strain transformed the peanut. However, the main clades are clearly identified.

The tree was created based in nucleotide sequences, since not all of cus homologs are intact. Thus, in  case of using amino acid sequences, we would lose a lot of data on В genome.

Not only IQTREE, but also PhyML  incorporates the possibility of testing which is the best model of sequence evolution. We have provided more detailed information on the phylogenetic analysis methods used, including model testing as follows:

 “Nucleotide sequences were aligned using the MUSCLE tool for multiple sequence alignment [25]. The evolutionary history was inferred using the maximum-likelihood method and the Kimura 1980 model [26] using PhyML [27] with smart model selection [28]. The operation of this tool included selecting the best model using the BIC criterion. The initial trees for heuristic search in the ML method were obtained automatically using the BioNJ algorithm. 100 transfser-bootstrap datasets [29] have been created. in Final tree was visualized in MEGA11. Branches corresponding to partitions reproduced in less than 50% bootstrap replicates were collapsed The percentage of replicate trees in which the associated taxa clustered together are shown next to the branches [30].“

The existence of alleles and phylogenetic relationships between them are different concepts.

Alleles exist physically, as states of specific genes. We can study their relationship using different methods. And we can trust our conclusions about the closer relationship of certain alleles, depending on the bootstrap values.

  1. R:The authors should include in the Introduction all the data concerning the different ancestral genomes. They describe the A and B genomes, but not the others (C, E, PR...) that appear later in the text.

C, E, PR are not ancestral genomes of cultivated peanuts. They are just relative. We mentioned them in the new version of the introduction.

  1. R: It would be very convenient to have a small figure showing the evolutionary relationships among the Arachis species discussed in the text. Otherwise, it is very difficult to follow the discussions on when the opine sequences appeared.

Opine sequences appeared before all analyzed species divergence (it is written at lines 177-178 ).   Unfortunately, the brief communication format does not allow adding another figure.

  1. R: Also, it would be convenient to show in detail the data concerning the precise chromosomal locations of the sequences. How did the authors establish whether they are exactly the same in different species? If indeed they are able to map the sequences (line 145), why is it then impossible to establish whether the sequences come from a single gene or two genes in tandem? (as suggested on line 178).

We discussed chromosomal locations on the basis of high-quality genomic assemblies. The gene under study is located on chromosome 8. In the case of assemblies from SRA reads, the same localization site can be assessed. However, the coverage in different samples was different, as follows from Table 1. Somewhere we could only say that such a gene exists. Somewhere we could assembly it to conclude whether it was full-sized, somewhere there was enough data for phasing alleles. Reconstruction of repeats is the most difficult task. Usually, long reads are needed to assemble repeats well. Our idea of ​​different copy numbers has a right to exist, since in WGS data the number of transgenes varies in different varieties of A. duranensis. We described it as follows:

«The variability of transgene copy number can be illustrated using genomic assemblies obtained before us for different varieties of the diploid species A. duranensis. Thus, there are 2 copies of the transgene in the genome assembly of cultivar PI475845. At the same time, each of cultivars K30060 and V14167 contain one copy of the transgene».

  1. R: Figure 1B: I do not understand why two alleles may come from a single branch in Figure 1A. Are they just replicates?. Also, it would be convenient to include in Figure 1B the names of the alleles, perhaps below the expression bars.

One allele is one branch. But several genotypes can have the same allele (2 asterisks next to the single branch).

New version of figure 1b contains names of the genotypes below the expression bars

  1. R: Were the first GMOs described in 2019? (line 47)

The first natural GMOs were described much earlier within the genus Nicotiana. In 2019, it became clear that peanuts are a natural GMO (clarified in the introduction)

Reviewer 2 Report

Comments and Suggestions for Authors

The authors conducted an in-depth, thorough study of the variability of cus-like gene in peanuts derived from T-DNA of Arobacterium. The article will undoubtedly be of interest to specialists studying naturally occurring transgenic plants.

There are some comments on the manuscript regarding the presentation of the data obtained.

Table S3 contains 94 sequences, most of which are from the SRA database and are designated as SPX.... The origin of the remaining data must be indicated, the sequences are named as B, C, A, E, D, ODRX231922_Arachis_cardenasii_A, 1157, 905.

The following Accession numbers not found: JAAOXE010000008.1_Arachis duranensis cv K30065, JAPZLN010000006.1_Agrobacterium salinitolerans strain, JQIO01000321.1:5264337-5265082 Arachis ipaensis.

 Section 3.2:

Lines 171-172; 184 - Not 28 but 27 cus-like gene alleles.

Line 173: change Table S3 to Table S4

Line 180:11 genotypes from  both subspecies” change to “12 genotypes from both A. hypogea subspecies (Table S4)”

Line 183: in three accessions > in three accessions each

Table S4: add A. duranensis to the table; Label columns E and G.

In Table S4, 19 sequences are obtained from the SRA database. We can only guess that the remaining 8 are results of Sanger sequencing of samples from the VIR collection. Of these, only 2 are deposited in the GenBank.

Lines 187-88: “which confirms the adequacy of the applied methods.” – what methods do you mean?

Lines 188-192: these conclusions do not follow from the data presented.

Line 201: “equidistant from clades A and B” – really, A. helodes sequences closer to clade B.

Figure 1: Expand the legends for the pictures

Figure 1B: What do the asterisks mean, what is the meaning of their names?

Lines 205-206: “The lower divergence of genome A sequences”  - this is not obvious from the presented tree since the values of genetic distances between alleles are not given.

Author Response

The authors express their deep gratitude to the reviewer for the careful reading of the manuscript, its high assessment, and valuable comments.

  1. Table S3 was corrected according to the reviewer`s comments.
  2. The following Accession numbers not found: JAAOXE010000008.1_Arachis duranensis cv K30065, JAPZLN010000006.1_Agrobacterium salinitolerans strain, JQIO01000321.1:5264337-5265082 Arachis ipaensis. These are WGS accession numbers. They are correct.
  3. Section 3.2:

Lines 171-172; 184 - Not 28 but 27 cus-like gene alleles.

There are 29 alleles

Unique alleles

  1. .H1
  2. .H2
  3. .H1
  4. .H2
  5. .H1
  6. .H2
  7. .H1
  8. .H2
  9. .H2
  10. .H1
  11. .H2
  12. .H2
  13. .H1
  14. .H2
  15. .H1
  16. .H1
  17. .H1
  18. .H2
  19. .H1
  20. .H1
  21. .H2
  22. 416/ 1
  23. 555/ OL840910.1
  24. 1157/ OR905615.1
  25. 1905/ OR905616.1

Common alleles

  1. A
  2. B
  3. С
  4. D

Line 173: change Table S3 to Table S4 - corrected

Line 180: “11 genotypes from  both subspecies” change to “12 genotypes from both A. hypogea subspecies (Table S4)” - corrected

Line 183: in three accessions > in three accessions each - corrected

Table S4: add A. duranensis to the table; Label columns E and G. - corrected

In Table S4, 19 sequences are obtained from the SRA database. We can only guess that the remaining 8 are results of Sanger sequencing of samples from the VIR collection. Of these, only 2 are deposited in the GenBank. – corrected.

All sequences were deposited in the Genbank (except line 1252, identical to others with allele A). Their numbers were provided in the Data Availability Statement section. In the new version of the manuscript they are also added to table S4.

Lines 187-88: “which confirms the adequacy of the applied methods.” – what methods do you mean?

Replaced by: “It is worth noting that we found common alleles in both the SRA assembly data and the Sanger sequencing results of samples from the VIR collection, indicating consistent results from these methods. ”.

Lines 188-192: these conclusions do not follow from the data presented.

Replaced with : ”The presence of the common intact alleles from the A genome in half of the studied genotypes of A. hypogaea and ancestral A. duranensis suggests that there is stabilizing selection in favor of the intact allele. However, such selection is absent for the alleles from the B genome, causing them to accumulate mutations independently. ”.

Line 201: “equidistant from clades A and B” – really, A. helodes sequences closer to clade B. -corrected

Figure 1: Expand the legends for the pictures - corrected

Figure 1B: What do the asterisks mean, what is the meaning of their names? – Different lines from VIR collection are shown with different colors. Asterisks of certain color indicate certain line from VIR collectionon the tree. Corrected

Lines 205-206: “The lower divergence of genome A sequences”  - this is not obvious from the presented tree since the values of genetic distances between alleles are not given. – replaced with “diversity”

Reviewer 3 Report

Comments and Suggestions for Authors

I don't know the purpose and the significance of this manuscript after thoroughly reading. Why use agropine synthesis gene and cucumopine gene to study the relationship of the Arachis? I suggest authors to clearly explain the two genes, GMO, and cT-DNA in the introduction part. The authors can use a flowchart or figure to help illustrate the project. Besides, the English writing of the manuscript is very hard to understand for some parts.

Comments on the Quality of English Language

Need to improve.

Author Response

The authors would like to express their gratitude to the reviewer for their time, valuable comments, and advice.
We have made revisions to the English language.
The introduction now contains more detailed information on agrobacteria, T-DNA, the genes within it, and natural GMOs.

“Agrobacterium-mediated transformation, now referred to as Rhizobium-mediated transformation, is a commonly used method for obtaining genetically modified crops. This involves the transfer of a fragment of the bacterium's plasmid, called T-DNA (transferred DNA), into the plant's chromosome. The T-DNA genes contain eukaryotic promoters that allow for their expression in the plant. This expression leads to the development of crown gall or hairy roots diseases, as well as the production of opines, which serve as a source of nutrition for the bacterium. Crown galls and hairy roots are transgenic tissues that form on non-transgenic plants. Over time, scientists have learned how to replace the internal T-DNA sequence with genes of interest, and these genes can be transferred into plant cells, resulting in the regeneration of whole transgenic plants [1].  At the same time, some plant species have undergone transformation by Agrobacterium or Rhizobium naturally, without any human involvement. They were firstly described inside the genus Nicotiana and are referred to as naturally transgenic or natural genetically modified organisms (nGMOs) [2-4]. These plants contain sequences in their genomes called cellular T-DNAs (cT-DNAs), which are homologous to "agrobacterial" T-DNAs. These cT-DNAs are passed down to their offspring from generation to generation [5]. Since 2019, many nGMOs have been identified among various crop species, and most of them contain genes for opine synthesis in their genomes [6-8].

The original rationale for using the cucumopine synthase gene to study the relationship of Arachis was already present in the text. We have chosen not to make any changes since our approach has not previously received any criticism in studies of the genera Camellia and Vaccinium.
The information is presented in the text as follows.

«Moreover, the examination of polymorphisms in recently acquired cT-DNA sequences can help to resolve contentious issues in peanut phylogeny. These sequences offer numerous advantages for investigating the origins and evolution of nGMO species. cT-DNAs are well-defined, highly specific, and easily identifiable DNA fragments that are distinct from plant sequences. Integration at a specific chromosomal site in the plant genome indicates a founder event, thereby establishing a clear starting point for a new clade. Additionally, cT-DNAs can be of sufficient size and age to produce a range of variations for constructing phylogenetic trees. By analyzing the evolutionary trajectory of transgenes in a closely related group of species, researchers can gain insights into how they evolve [15]. Phasing T-DNA alleles allows for the reconstruction of more accurate phylogenetic relationships and the detection of hybridization events during plant species evolution. These methods have been effectively applied in studying the genera Camellia and Vaccinium [17, 18]».

Furthermore, we have included an additional sentence to the Discussion: “

Further research necessitates the inclusion of additional lines, varieties, and species of Arachis. When examining a vast number of samples from different species, it is essential to employ cost-effective and efficient research methods for phylogenetic analyses. In addition to other techniques, incorporating a molecular marker based on a recently evolved sequence can provide a more comprehensive understanding of the phylogeny of Arachis.

.”

Round 2

Reviewer 1 Report

Comments and Suggestions for Authors

Review of “Evolutionary fateofthe opine synthesis genes in the Arachis L. genomes” by O. D. Bogomaz et al.

The revised version is much better than the first draft. Language has been much improved and several questions answered in the text. However, there are still some problems, which I think may be easily solved by the authors.

Major questions:

1)      It is unclear whether the classification into alleles is supported by the data. From Figure 1, it can be deduced that there is a large variation in the cus-like sequences. I understand that each one of the lines in the tree is an allele (although some sequences apparently may be identical). I do not see how the classification into alleles in the text fits with that figure. In the text, alleles are classified without considering the tree. So, what exactly is, for example, “Allele A” is unclear. If it corresponds to all sequences that are included in what later is called “Clade A”, this should be explained in exactly the opposite way, that is, the tree should be described first and only then alleles defined.

2)      Definition of groups of alleles thus depends on the tree shown in Figure 1. Now, please consider that groups with bootstrap support below 70% are not to be trusted (as shown long ago by Guindon et al. Syst. Biol. 59:307-321 [2010] and several other papers). So, for example, clade C has no support. On the other hand, it could be better to include several sequences into clades A and B which now are kept separated, but are adjacent in the tree, put together in the tree in branches with strong statistical support.

Minor question:

1)      The closeness of A. diogoi and A. hoehnei sequences in two places of the tree is attributed to hybridization. Is there any proof of them not being simply gene duplicates in two close species?

Some suggestions (lines refer to the version with the original first draft plus corrections)

Simple summary, line 21: “having been acquired through stabilizing selection” substitute by “probably being under a regime of stabilizing selection”

Introduction, line 68. “contain genomes A and B” perhaps better “ancestral genomes known as A and B”

Intro, line 85: “including cultivated Arachis”, change to “among cultivated Arachis”

Intro, line 98; “range for variations for constructing”, change to “range of variants, which allow constructing…”

Methods, line 172: “ using the MUSCLE tool for multiple…”, change to “using MUSCLE”. Also, it would be good to say that the parameters used were the ones that the program offers by default.

Methods, line 177. “100 transfer bootstrap datasets”, change to “To test the support for the topology of the tree, 100 boostrap replicates were obtained”.

Methods, line 177: “final tree” – “the final tree”

Results, line 189: “A. ipaensis, 1 accession each of…” – “A. ipaensis, and, finally, 1 accesion each of…”

Results, line 197: “Our results” – “In summary, our results…”

Results, line 204: “Culorrhizae, Heteranthae” – “Culorrhizae and Heteranthae”

Results, line 214: “previosly described procedures” – “procedures previously described”

Results, line 227: “assemblies obtained before us” – the meaning of this phrase is unclear

Results, line 248: “mutations independently” – “mutations”

Discussion, line 316: “began to mutate” – “accumulated mutations”

Comments on the Quality of English Language

/

Author Response

We would like to express our gratitude to the reviewer for thoroughly reviewing the manuscript and providing valuable comments.

Please find our responses to these comments presented below.

Major questions:

  • It is unclear whether the classification into alleles is supported by the data. From Figure 1, it can be deduced that there is a large variation in the cus-like sequences. I understand that each one of the lines in the tree is an allele (although some sequences apparently may be identical). I do not see how the classification into alleles in the text fits with that figure. In the text, alleles are classified without considering the tree. So, what exactly is, for example, “Allele A” is unclear. If it corresponds to all sequences that are included in what later is called “Clade A”, this should be explained in exactly the opposite way, that is, the tree should be described first and only then alleles defined.

Answer

An allele is one of the different forms of a gene. In diploid organisms, we can expect the presence of two alleles for each gene (heterozygous state). However, most whole genome sequencing (WGS) assemblies currently represent an average variant between the two alleles. Consequently, we lose valuable information about potential hybridization events, which can lead to errors when establishing phylogenetic relationships.

To address this issue, we first identified the specific alleles and then utilized them to construct the phylogenetic tree. This sequence of events was initially outlined in the materials and methods section.

In the current version, we have made some clarifications to simplify the text. To prevent any confusion between alleles and clades, we have used Greek letters to label the clades on the tree, while the alleles are named using Latin.

2)      Definition of groups of alleles thus depends on the tree shown in Figure 1. Now, please consider that groups with bootstrap support below 70% are not to be trusted (as shown long ago by Guindon et al. Syst. Biol. 59:307-321 [2010] and several other papers). So, for example, clade C has no support. On the other hand, it could be better to include several sequences into clades A and B which now are kept separated, but are adjacent in the tree, put together in the tree in branches with strong statistical support.

Answer

 The phylogenetic tree has been corrected according to comments.

All recommended corrections have been made to the text. 

Reviewer 3 Report

Comments and Suggestions for Authors

The purpose of this research seems to examine the presence of cT-DNA in the Arachis, however, this is not necessarily related to GMO. Besides,  a lot of researches about peanut phylogeny have been done using different molecular markers, such as SNPs, SSRs. The evolutionary relationship of Arachis is almost clear now. I don't know why use the cT-DNA to study the phylogeny. Overall, the paper lacks of logic and the significance is small.

Comments on the Quality of English Language

The English writing need to be improved. 

Author Response

We thank the reviewer for their time.

As we understand, the sections of the article dedicated to describing the diversity of natural transgenes in Arachis and assessing their expression did not receive any comments.

The question only arose in the section concerning the construction of a phylogeny based on a natural transgene. We are aware of articles that used microsatellites, ITS, chloroplast sequences, and even whole-genome data to reconstruct phylogenies. These works are cited in the text. Currently, the origin of cultivated peanuts is not a matter of controversy, but the phylogeny of the genus as a whole is. We are not the only ones to notice that the data from different studies are contradictory. Therefore, different approaches are needed to obtain a complete picture. Among these approaches, our method has a valid place, as it is affordable and less labor-intensive.

We describe all of this in the text.

It is well accepted that A. duranensis and A. ipaensis  are the progenitor species of A. hypogaea [36, 39]. In recent years, the scientific community has accepted the idea of a common origin for A. hypogaea and A. monticola. However, there is no complete clarity regarding the rank of differences between them. Yin et al., based on a comparative analysis of NGS data, proposed that wild tetraploid A. monticola was formed by hybridization between two diploid species A. duranensis and A. ipaensis and then evolved into A. hypogaea by artificial selection [40]. Tian and coauthors [38] reported that A. monticola could be a weedy subspecies of cultivated peanuts, since A. hypogaea is able to hybridize with A. monticola and produce fertile hybrids. Thus, A. monticola may have been derived from a more ancient hybridization event, while the accessions of A. hypogaea may have evolved later. Our data are more consistent with the second hypothesis because the alternation of sequences of A. hypogaea and A. monticola within one clade on the tree is most logical to explain through hybridization events, and therefore the absence of a complete interspecific barrier.

There are several studies showing that cultivated peanuts may have been derived from more than two progenitor species, including such species as A. diogoi,  A. cardenasii, A. batizocoi in addition to well-accepted ones [41-45]. Within our study, there is no evidence in favor of the participation of mentioned above species in the formation of cultivated peanuts. However, it cannot be ruled out that the genetic material of A. paraguariensis was involved in the formation of at least some varieties of A. hypogaea. Data from other researchers support the close relationship between A. paraguariensis and A. duranensis [46]. The location of other species on the tree does not agree well with the classical division of the genus into sections. For example, some alleles of A. diogoi and A. hoechnei form a common clade, while others are located outside of this clade. A similar pattern was previously observed by Tian and co-authors, who studied chloroplast sequences of Arachis species [38]. This can be explained by hybridization in the context of incomplete speciation.